# Uncertainty-Aware Hierarchical Refinement for Incremental Implicitly-Refined Classification

**Jian Yang**[1][*]  **Kai Zhu**[1,*,‡]  **Kecheng Zheng**[2]  **Yang Cao**[1,3,†]

[1] University of Science and Technology of China  [2] Ant Group

[3] Institute of Artificial Intelligence, Hefei Comprehensive National Science Center

{yangjian12138@mail, zkzy@mail}.ustc.edu.cn

zkccloud@gmail.com  forrest@ustc.edu.cn

## Abstract

Incremental implicitly-refined classification task aims at assigning hierarchical labels to each sample encountered at different phases. Existing methods tend to fail in generating hierarchy-invariant descriptors when the novel classes are inherited from the old ones. To address the issue, this paper, which explores the inheritance relations in the process of multi-level semantic increment, proposes an Uncertainty-Aware Hierarchical Refinement (UAHR) scheme. Specifically, our proposed scheme consists of a global representation extension strategy that enhances the discrimination of incremental representation by widening the corresponding margin distance, and a hierarchical distribution alignment strategy that refines the distillation process by explicitly determining the inheritance relationship of the incremental class. Particularly, the shifting subclasses are corrected under the guidance of hierarchical uncertainty, ensuring the consistency of the homogeneous features. Extensive experiments on widely used benchmarks (*i.e.*, IIRC-CIFAR, IIRC-ImageNet-lite, IIRC-ImageNet-Subset, and IIRC-ImageNet-full) demonstrate the superiority of our proposed method over the state-of-the-art approaches.

## 1 Introduction

In recent years, deep learning has made huge breakthroughs in the field of computer vision, matching or even surpassing human performance on some image recognition tasks [3]. However, learning multiple tasks [25] in sequential data (*i.e.*, continual learning [20, 22]) remains a major challenge, which requires models to have the ability to aggregate different learning objectives into a coherent whole over time.

When a new task comes with the increase of identified classes (*i.e.*, class-incremental learning), joint training with all old and new data is too time-consuming and labor-intensive. Furthermore, most data from past tasks are unavailable due to the data privacy. To adapt rapidly to the new scenarios, previous methods [26] try to adopt a simple alternative that directly fine-tune the network with new data. However, this may severely degrade the performance of the old class due to the bias of the feature extractor and classifier towards the new class [15], which is also known as catastrophic forgetting.

To address the issue, existing methods [16] maintain the performance of the old class by preserving the representative samples (*i.e.*, exemplar) and aligning the output distribution (*i.e.*, knowledge dis-

---

[*]Co-first Author

[†]Corresponding Author

[‡]Work done during an internship at Ant Group

36th Conference on Neural Information Processing Systems (NeurIPS 2022).

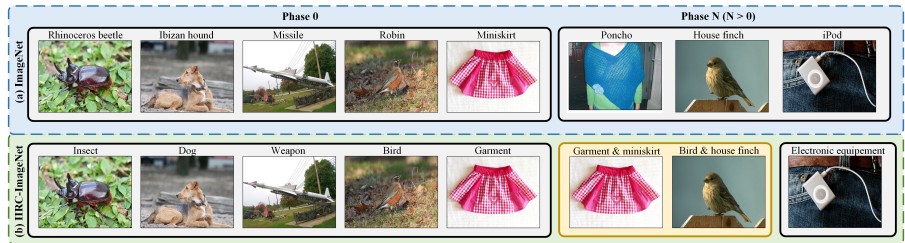

Figure 1: The setting of IIRC. Compared with the standard incremental process, superclasses and subclasses under the IIRC setting are mixed in the incremental process, and the labels of subclasses images are refined progressively, even for the same image.

tillation). This joint optimization with incremental samples strikes a balance between stability and plasticity, resulting in better predictions that fall into the label space of the old or new class.

**However, is incremental learning always a binary classification option that distinguishes between the old and new class?** In existing incremental learning settings (*e.g.*, ImageNet in Fig. 1(a)), each instance is arranged in a certain incremental stage in an exclusive manner. However, in real life, people's semantic understanding of the same instance may be gradually enriched as the learning process proceeds. As shown in Fig. 1(b), the semantics of the same image is refined from Garment to miniskirt in the IIRC-ImageNet setting. This paper focuses on the incremental ability to progressively learn and maintain multi-level semantic information, which is also known as Incremental Implicitly-Refined Classification (*i.e.* IIRC) [1].

**How to discern the semantic inheritance relationship in a hierarchical incremental scenery?** We find that the incremental performance for multi-level recognition decreases dramatically when applying classical class-incremental learning methods directly to the IIRC setting. Our analysis is mainly attributed to the following reasons. As shown in Fig. 2, on the one hand, although the probability distributions of classes with inheritance relationships show an obvious consistency in the initial phase, the incremental subclass inherited from a certain old class gradually outgroups under the supervision of new labels. It destroys the integrity of the representation of the whole old class, thus losing the semantic relevance of the hierarchical labels. On the other hand, some incremental classes inherit from none of the existing classes, leading to feature confusion due to the lack of old supervision.

To this end, we propose an Uncertainty-Aware Hierarchical Refinement (UAHR) scheme, which exploits the correlation of hierarchical distributions to guide the optimization of incremental representation. Concretely, a global representation extension strategy is proposed to widen the distribution distance among all new classes in the embedding space, enhancing their discriminative properties. Furthermore, a hierarchical distribution alignment strategy is further proposed to correct the optimization of the shifting subclasses by aligning with the distribution of the whole superclass, ensuring the consistency of the hierarchical uncertainty. In this way, we use RBF mapping to explicitly measure the distance in the feature space between the training samples and the class centers, quantifying the feature correlation. In the incremental phase, we calculate the entropy distribution of new classes in the old embedding space for the estimate of hierarchical uncertainty. The resulting differences and similarities are utilized to identify the multi-level semantic relationships, refining the subsequent distillation objectives. Comprehensive experimental results on IIRC-CIFAR, IIRC-ImageNet-lite, and IIRC-ImageNet-full datasets demonstrate the superiority of our method.

## 2 Related Work

### 2.1 Class-Incremental Learning

Existing class incremental learning methods can be mainly divided into four types. **Regularization-based** methods preserve high-weight parameters by estimating the importance of individual model parameters [2, 12, 24], while allowing unimportant parameters to be updated flexibly to ensure the learning ability for new classes. However, in the study of [20], it is shown that such methods have poor generalization performance in class incremental learning.

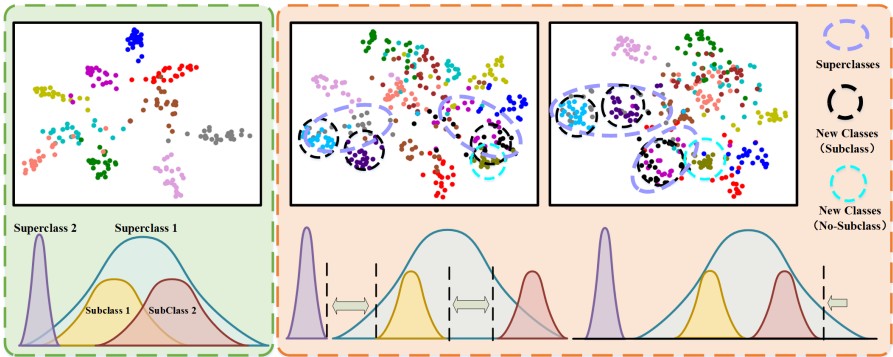

**(A) Initial phase**  **(B) Incremental phase**

Figure 2: Motivation of our method. The class with inheritance relationships has consistency in the initial distribution. As incremental learning proceeds, the subclasses shift away from the distribution of the corresponding superclasses and mix with new classes without subclasses.

**Distillation-based** methods encourage the model to learn new tasks, while the model representation obtained by the current training data is forced to mimic the representation of the old model. The new model in [14] completes knowledge distillation by matching the output of the sigmoid function with the old model. LUCIR [10] mitigates forgetting by distilling the output of the softmax function with the temperature scale and L2 regularization. LwM [7] reduces forgetting by matching gradient attention maps from the highest-scoring labels. PODNet [8] performs knowledge distillation between old and new models by retaining intermediate features pooled along width and height, and controls the balance between previous knowledge and new information, resulting in a more adaptive incremental representation.

**Rehearsal-based** methods work by storing a limited number of representative examples or adopting a generative model for old class samples when training new tasks. Incremental Classifier Representation Learning (iCaRL) [16] retains a small number of samples for each class to approximate class centroids and make predictions based on the nearest class average classifier. [18] generates previously observed class samples by using generative adversarial networks (GANS) [9]. **Structure-based** methods [27] keep the learned parameters related to the previous classes unchanged, and assign new parameters in different forms, such as unused parameters, extra networks, etc., to learn new knowledge. DER [23] concatenates the extracted features by adding a new feature extractor in the incremental phase. A sparsity error is adopted to encourage the model to compress parameters as much as possible. Different from the existing incremental methods focusing on the invariant semantic concepts, our method is designed for the progressive understanding of the incremental semantic information, facilitating better feature update and retention.

## 3 Our Method

### 3.1 Problem Description and Analysis

In real life, the human brain gradually enriches the semantic understanding of an object as the learning process advances. For example, we only know that rats are small animals in the young phase, but we understand that rats are rodents as we grow up. To simulate this process, we follow the incremental implicitly-refined classification task (IIRC), which is arranged into $N$ phases in the order of data stream with hierarchical labels, denoted as $\{T_1, \cdots, T_N\}$. Different from the traditional incremental learning settings, the samples of incremental phases in IIRC may have one or two labels (*i.e.,* hierarchical labels). Specifically, the hierarchical labels refer to that the one label of a sample is a subclass (*i.e.,* hamster), and the other one is a superclass (*i.e.,* small animal). The number of samples with superclass labels is always larger than that with subclass labels, which increases with the number of subclasses. During training, we always follow an incomplete information setup, *i.e.*, samples with multiple labels are only provided with the label in the current phase. We only use a subset of all superclasses in the first phase to train the model. Subsequent phases contain a mixture of superclasses and subclasses.

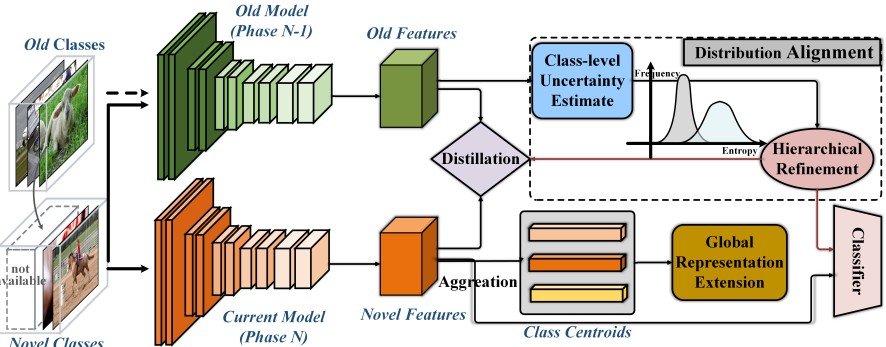

Figure 3: Our proposed Uncertainty-Aware Hierarchical Refinement scheme for IIRC.

We face two challenging problems in this setting: (1) How to judge the relationship among labels across phases, especially the superclass-subclass relationship? Moreover, how to guide the optimization of the novel class by means of the distribution relationship with superclass exactly? (2) How do we reduce the feature confusion among these novel classes in the incremental phase? Therefore, We propose two strategies. (1) Global Representation Extension. We use the RBF kernel to calculate the uncertainty among all class centers, which is minimized for training during the incremental phase. (2) Hierarchical Distribution Alignment. We firstly calculate the output entropy of the new class samples with the old model, estimating the uncertainty of the classes across phases. Then we use the obtained uncertainty to discriminate the superclass-subclass relationships and other relationships. Finally, we update the old model's output to guide the new model's optimization direction. Thus, we can keep the distribution invariance between the subclass and the superclass.

## 3.2 Global Representation Extension

Optimization with only the data of new classes during the incremental phase causes great bias in the representation and classifier learning. To maintain the stability of the feature space, we use the representation distance as a measure of uncertainty. New class representation is optimized by comparing the output entropy with the target labels. At the same time, the stability of the old class representation is ensured by maintaining the same output between the old and new models with the new samples. Specifically, we use the RBF distance [19] to calculate the representation extension loss. We use the RBF kernel to map representation to infinite dimensional space for comparison, which is shown as follows,

$$\mathcal{L}_{div} = \sum_{c=0}^{n_b} K(h_\theta(\boldsymbol{x})_c, h_\theta(\boldsymbol{x})_{j_{near}}) = \sum_{c=0}^{n_b} \exp\left[\frac{-\frac{1}{n_d}\|h_\theta(\boldsymbol{x})_c - h_\theta(\boldsymbol{x})_j\|_2^2}{2\sigma^2}\right], \quad (1)$$

where $K(\cdot)$ denotes the Radial Basis Function kernel (RBF kernel [11]), $n_b$ denotes the number of classes contained in the current batch, $h_\theta(\cdot)$ denotes the backbone of our net, we can get the output features of the picture from it, $n_d$ denotes the output dimension after global pooling, and $h_\theta(\cdot)_c$ denotes the average feature value of a particular class $c$. The symbol $j$ represents the class number with the smallest distance to class $c$ in the feature space, which is not equal to $c$. The optimization of this loss forces the extension of the distance among the class centers, obtaining better discrimination in the class distribution. Subsequently, we use a linear layer to get the entropy of the samples, then judge the class of these samples. We adopt the BCEWithLogitsLoss to optimize the learning, which is shown as follows,

$$\mathcal{L}_{cls} = -\frac{1}{N}\sum_{i=1}^{N}[y_i \log(\text{sigmoid}(f_\theta(h_\theta(\boldsymbol{x}_i)))) + (1-y_i)\log(1-\text{sigmoid}(f_\theta(h_\theta(\boldsymbol{x}_i))))], \quad (2)$$

where $y_i$ denotes the ground truth label of the sample $i$, $N$ denotes the batch size of training, $x_i$ denotes the $i_{th}$ sample, and $f_\theta(\cdot)$ denotes the classifier of the current model. It should be noted that the labels here as well as the final output of the model contain only the classes of the current phase. The optimization of the $loss$ allows the model to eventually judge the label of a sample by whether

---

**Algorithm 1** Acquisition of Hierarchical Semantic Relationship

---

**Input:** The outputs of the samples of class $c$ on old model $output$, the number of samples $height$, the number of classes for each phase $Stage$, the global pooled features of the sample of class $c$ on old model $feat$, and the number of the current phase $idx$.

1: Declare an initial label matrix $M_{ori}$ of size $(height, idx)$.
2: **for** each $j \in [0, height]$ **do**
3:     **for** each $k \in [0, idx]$ **do**
4:         $S_{max}, L_{max} = \max(output[j, Stage[k]{:}Stage[k+1]])$
5:         $M_{ori}[j, k] = L_{max}$
6:     **end for**
7: **end for**
8: Declare a phase label statistics matrix $M_{pha}$ of size $(2, idx)$.
9: **for** each $j \in [0, idx]$ **do**
10:     $L_{pha}, C_{pha} = \text{countmax}(M_{ori}[:, j])$
11:     **if** $L_{pha} < height \,/\, 2$ **then**
12:         $L_{pha}$ = -1
13:     **end if**
14:     $M_{pha}[0, j] = L_{pha}$
15:     $M_{pha}[1, j] = C_{pha}$
16: **end for**
17: For all labels in the current $M_{pha}$ that are not -1, an equal number of corresponding sample features are randomly extracted from $feat$, and the standard deviation is calculated for each, forming $Std_{pha}$.
18: Calculate the standard deviation std for the current $feat$, declared as $Std_c$.
19: $Super = \text{argmin}(|Std_{pha} - Std_c|)$

**Output:** the superclass of current class C : $Super$.

---

its output is greater than 0. And the closer the corresponding representation is to the class center, the larger the output value. Finally, we use the output value of the new sample on the old model as the supervised signal of the distillation process. Thus, the current model learns new classes while maintaining the knowledge of the old model, which is shown below:

$$\mathcal{L}_{dis} = \text{BCEWithLogitsLoss}(f_\theta(h_\theta(\boldsymbol{x}))[:, : n_{old}], f_\theta^*(h_\theta^*(\boldsymbol{x})/\alpha)), \tag{3}$$

where $h_\theta(\cdot)$ denotes the backbone of the current model, $f_\theta(\cdot)$ denotes the classifier of the current model, $h_\theta^*(\cdot)$ denotes the backbone of the old model, $f_\theta^*(\cdot)$ denotes the classifier head of the old model, $\alpha$ is a scale hyperparameter of distillation, and $n_{old}$ denotes the number of classes that have been learned. $\boldsymbol{x}$ is a training sample from the current phase. The optimization of this $loss$ facilitates the stability of the old class feature distribution.

### 3.3 Hierarchical Distribution Alignment

In the incremental phase, focusing on the label relationships is the key to solving the IIRC problem. Firstly, the new subclass samples can be mapped into the embedding space by the old model with fixed parameters, which is close to the corresponding superclass distribution. Then the label relationships can be inferred from the hierarchical uncertainty, which is calculated by counting the entropy of new samples on the old model across phases. At the same time, we flexibly adjust the distribution border to correct the shift of subclass samples and mitigate the confusion caused by the brother class.

Specifically, as shown in Fig. 3, we take the new class $C$ as an example. Firstly, we obtain all mapped features and corresponding output entropy of each old class on the old model with the new samples of class $C$. Then, according to the obtained output entropy, the ratio of each old class distribution is calculated. This ratio is the hierarchical uncertainty of the current added class $C$ samples to each old class label. The obtained hierarchical uncertainty is binarized to determine the corresponding old class with a similar distribution (possibly superclass-subclass or brother class relationship) to the current specific class $C$. Finally, We calculate the standard deviation of each similar old class with the same operation. Direct superclass-subclass relationships have closer standard deviation values on the feature distributions. So the similar old class with the closest std value is judged as

| GRE | HDA | IIRC-CIFAR | | | | |
|-----|-----|---------|---------|----------|----------|----------|
| | | phase 0 | phase 5 | phase 10 | phase 15 | phase 21 |
| | | 78.35 | 26.48 | 21.27 | 18.81 | 17.78 |
| $\checkmark$ | | 77.04 | 26.75 | 21.46 | 19.38 | 18.32 |
| | $\checkmark$ | 77.06 | 29.31 | 24.73 | 22.42 | 18.38 |
| $\checkmark$ | $\checkmark$ | 77.53 | 30.11 | 25.31 | 23.56 | 19.05 |

Table 1: Ablation study of our method on IIRC-CIFAR.

the superclass of the current class $C$, and the rest of the classes are brother classes. The detailed procedure is shown in Algorithm 1.

After obtaining the label relationships, we use the selective distribution alignment distillation mechanism to guide the representation learning of models. According to the labeling relationship, we can achieve the alignment of the distribution and mitigate the confusion caused by the brother class. On one hand, when the features of the novel sample have a superclass shift, its output entropy value of the corresponding superclass on the old model is improved by a margin distance. Thus, the shifting samples are guided within the corresponding superclass distribution. On the other hand, the output entropy value of a novel sample that lacks a superclass but falls within the old class distribution is subtracted by a margin distance. Specifically, according to the label relationships, when the current new class C is not related to an old class, the highest output entropy by the sample of class C on the old model is subtracted by a margin value. Moreover, when the current new class C is a subclass of an old class, the output entropy value of the class C sample on the old model, which corresponds to the superclass, is added with one margin value. At the same time, the highest output entropy value of the non-superclass is subtracted by a margin value. Our selective distribution alignment distillation loss can be obtained by:

$$\mathcal{L}_{dis} = \text{BCEWithLogitsLoss}(f_\theta(h_\theta(\boldsymbol{x}))[:,:n_{old}], y^{new}),  \quad (4)$$

where $y^{new}$ is the new output of the old model, after performing our hierarchical distribution alignment strategy. In the end, our complete loss is:

$$\mathcal{L}_{all} = \mathcal{L}_{cls} + \mathcal{L}_{dis} + \mathcal{L}_{div} * \gamma,  \quad (5)$$

where $\gamma$ denotes hyper-parameters for balancing the losses. Moreover, in our experiments, $\gamma$ is set as 10.0. A detailed description of the hyperparameter selection is shown in supplementary material B.2.

## 4 Experiments

### 4.1 Dataset and Settings

According to the semantic relevance among labels, CIFAR100 [13] and ImageNet [6] datasets are rearranged to form the two-level hierarchy datasets [1]. Each label starts as a leaf node (i.e., subclass), and similar labels are assigned a common parent node (i.e., superclass). The integrated datasets are called IIRC-CIFAR and IIRC-ImageNet-full. **IIRC-CIFAR.** Ten superclasses are set up, each with about 4 to 8 subclasses. In incremental phases, each new phase introduces five classes. IIRC-CIFAR involves 22 phases with ten preset class orders called phase configuration for multiple tests. **IIRC-ImageNet-full.** In IIRC-ImageNet-full, sixty-three superclasses are set up, and the number of subclasses that belong to one superclass varies greatly, from 3 to 118. There are a total of 35 phases, with 30 classes per phase. Five preset class orders are fixed for multiple tests. **IIRC-ImageNet-lite** is a shorter, lighter version with just ten phases (with five task configurations), which is referred as IIRC-ImageNet together with IIRC-ImageNet-full. **IIRC-ImageNet-Subset** [21], as a simplified version of the IIRC-ImageNet-full, involves ten superclasses and 100 subclasses.

The superclasses of each dataset are combined by extracting 40% samples from each subclass, while the corresponding subclasses are saved with 80% samples. That means that the superclasses and subclasses share 20% samples. When the superclass contains more than eight subclasses, the samples of the subclasses are extracted in the proportion of $\frac{8}{number\ of\ subclasses} * 40\%$. The training process follows the incomplete information setting, *i.e.*, if a sample has more than one label, only the label of the current phase is provided. The validation set also follows the incomplete information setting.

| Method | IIRC-CIFAR | | | |
|---|---|---|---|---|
| | phase 5 | phase 10 | phase 15 | phase 21 |
| *Common Uncertainty* | 25.70 | 20.89 | 19.55 | 15.32 |
| *Ours* | 30.11 | 25.31 | 23.56 | 19.05 |

Table 2: Comparison to common uncertainty method on IIRC-CIFAR.

After a certain number of training phases, the overall performance is evaluated on a validation set and a test set under the complete information setting, which contains all seen class labels.

**Metrics.** The evaluation metric uses the precision-weighted Jaccard similarity (PW-JS) to measure the performance of the model on phase k after training on phase j, as follows:

$$R_{jk} = \frac{1}{n_k} \sum_{i=1}^{n_k} \frac{|Y_{ki} \cap \widehat{Y}_{ki}|}{|Y_{ki} \cup \widehat{Y}_{ki}|} \times \frac{|Y_{ki} \cap \widehat{Y}_{ki}|}{|\widehat{Y}_{ki}|}, \tag{6}$$

In the formula $j \geq k$, $\widehat{Y}_{ki}$ is the prediction value of the model for $i_{th}$ samples of $k_{th}$ phase, $Y_{ki}$ is the ground truth, and $n_k$ is the number of samples in this phase. Moreover, for the evaluation of the overall performance, we use the average precision-weighted Jaccard similarity of the model after training on phase j for all seen classes, as follows:

$$R_{j} = \frac{1}{n} \sum_{i=1}^{n} \frac{|Y_{i} \cap \widehat{Y}_{i}|}{|Y_{i} \cup \widehat{Y}_{i}|} \times \frac{|Y_{i} \cap \widehat{Y}_{i}|}{|\widehat{Y}_{i}|}. \tag{7}$$

## 4.2 Ablation Study

To prove the effectiveness of our method, we conduct a set of ablation experiments on the IIRC-CIFAR dataset. One of the essential components in our scheme is the global representation extension strategy, referred to as the GRE component. The other is the hierarchical distribution alignment strategy, referred to as the HDA component. To verify the functionality of the GRE and HDA components, we conducted experiments in the case of phase configuration 0 of IIRC-CIFAR. As shown in Table 1, the HDA component alone achieves an improvement of 0.6% in the last phase, while the GRE component alone achieves 0.54% in the last phase. Furthermore, with both components added, a 1.27% increase over baseline is achieved. It proves that both components play a positive role in evaluating the overall incremental performance.

In Fig. 6 (C), we show the incremental performance at each phase, which is evaluated over the test samples at a specific phase j, after training on that phase($R_{jj}$ using Equation 6). Such cure graphs evaluate the learning ability for new classes. We can see that the HDA component significantly improves the average performance while the GRE component further enhances the learning ability on top of the HDA component.

## 4.3 Analysis

**The impact of global representation extension.** To explore the impact of the global representation extension strategy, we visualize the t-SNE features of the baseline and our method. We select data from phase 0 and phase 4 of IIRC-CIFAR with phase configuration 0. In the upper part of Fig. 4, we can observe an overlap among the representation distributions of the new classes in the baseline method. Our method weakens the overlap and expands the distribution distance among the novel class. Fig. 4 proves that the usage of representation distance for the uncertainty estimate is adequate. Reducing this uncertainty leads to the distance extension between the individual class centers.

**The impact of the hierarchical uncertainty.** To explore the effect of hierarchical uncertainty, we first compare different uncertainty methods. A detailed version of Table 2 is presented in A.3 of the supplementary materials. As shown in Table. 2, the "Common Uncertainty" denotes an entropy-based uncertainty method widely used in the OOD [5]. We use it to construct label relationships. It can be seen from the Table. 2 that our method achieves better experimental results, proving that the entropy output and the standard deviation of the features exhibit the most significant impact on the cross-hierarchical uncertainty estimation among the statistical properties of the superclass and the corresponding subclass.

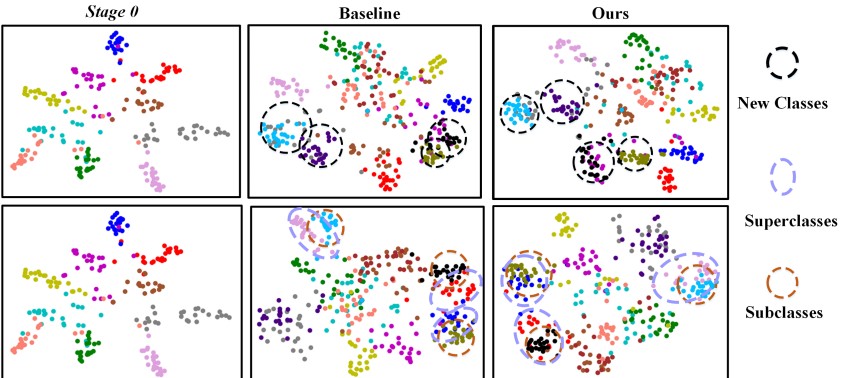

Figure 4: The impact of our method on the representations. (1) The upper row shows the effect of extending the distribution distance between the incremental classes. (2) The lower row shows the effect of the hierarchical distribution alignment.

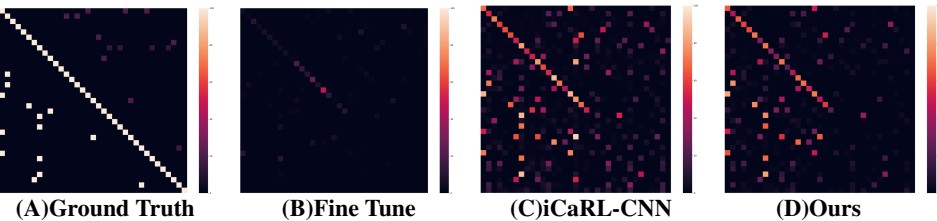

(A)Ground Truth        (B)Fine Tune        (C)iCaRL-CNN        (D)Ours

Figure 5: Confusion matrix for different methods.

To explore the effect of the hierarchical distribution alignment strategy, we visualize the t-SNE features for the baseline and our method on the IIRC-CIFAR dataset with phase configuration 0 by selecting data from phase 0 and phase 11. In the bottom half of Fig. 4, we can see a significant shift between the distributions of subclasses and corresponding superclasses in the baseline. While our method extensively corrects this shift and promotes the distribution of the novel subclasses falling in the corresponding superclasses. It demonstrates that adjusting the optimization direction of the feature distribution according to the labeling relationship helps the new model to learn better the representation relationship between the old and new classes and deepen it gradually.

## 4.4 Comparison with SOTA

To comprehensively measure the performance of our method under IIRC setting, we reproduces the classical methods (*i.e.*, ER [17], AGEM [4], LUCIR [10], iCaRL-CNN [16], iCaRL-norm [16], Podnet [8], HCV [21]) and compare with them.

**Average PW-JS.** It can be seen in Fig. 6 (A) that our method achieves the best until the end among all methods with incomplete information in IIRC-CIFAR. As shown in Fig. 6 (D), our method also maintains the highest scores in larger IIRC-Imagenet-lite, demonstrating the effectiveness of our method. As shown in Fig. 6 (E), our method achieves the SOTA result on the IIRC-ImageNet-Subset dataset. It can be shown in Fig. 6(F) that our method finally achieves the highest score on IIRC-ImageNet-full. More details are shown in A.5 of the supplementary materials.

**Performance on new classes.** The PW-JS values indicating the performance of the newly incremental phase are shown in Fig. 6(B). It can be seen that the ER algorithm and the AGM algorithm are superior in learning new classes. Combine with the performance in overall phases, they obtain an advantage in the new classes at the expense of the performance in old classes. Among all beneficial algorithms for suppressing catastrophic forgetting, our method achieves the best and outperforms existing methods in learning new classes.

We conduct an additional experimental set on the confusion matrix, where we extract the data from the IIRC-CIFAR with phases 0, 1, 2, 7, 12, and 17. They are combined and presented in Fig. 5. In the IIRC task, the diagonal values in the confusion matrix represent the prediction of subclass

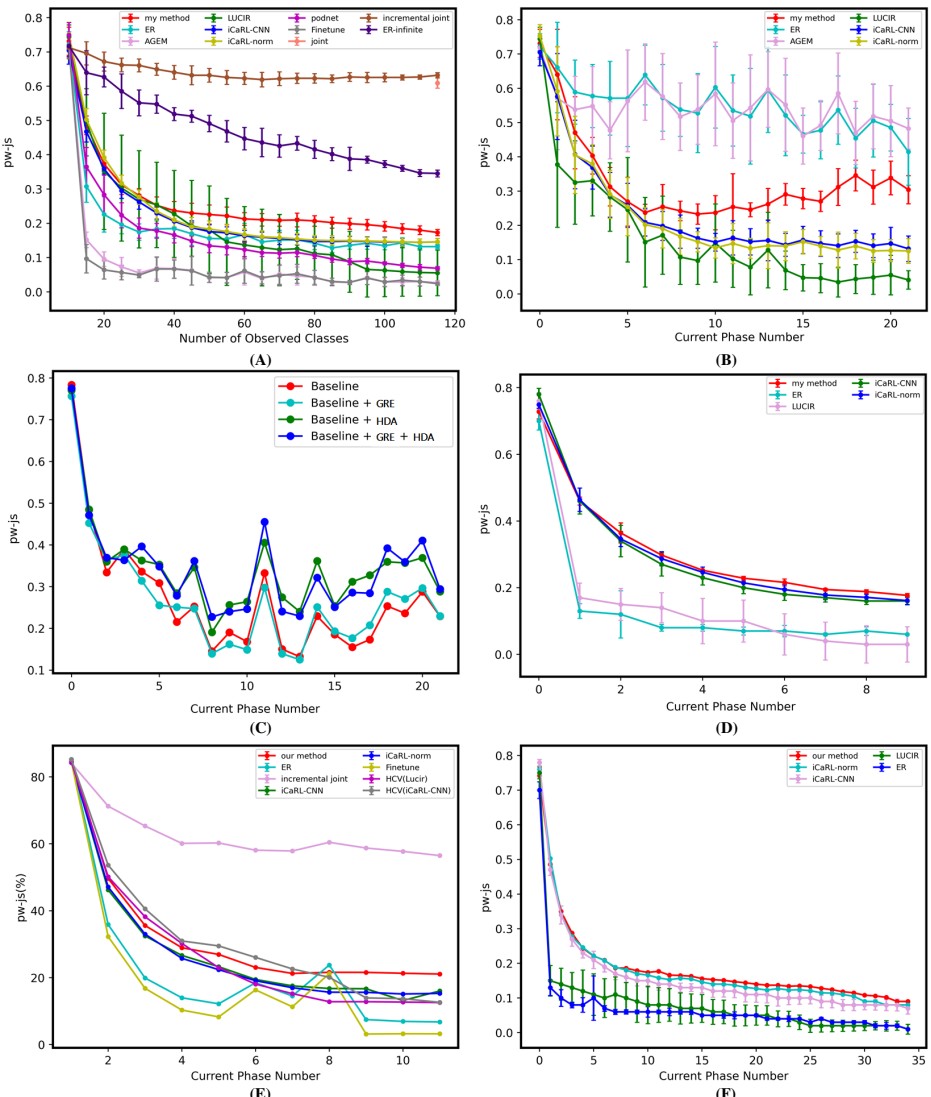

Figure 6: Curve graphs of the comparison experiments. (A) The overall performance comparison on the IIRC-CIFAR dataset. (B) The performance of the incremental phase on the IIRC-CIFAR dataset. (C) It shows the performance of two components of our method on the evaluation of new class learning ability. (D) The performance comparison on the IIRC-ImageNet-lite dataset.(E)The performance comparison on the IIRC-ImageNet-Subset dataset. (F)The performance comparison on the IIRC-ImageNet-Full dataset. ∗ indicates our re-implementation.

labels, while the bottom-left corner represents one of the superclasses. A more precise confusion matrix in the lower left corner represents better label relationship maintenance between subclass and superclass. As shown in Fig. 5, our method performs best in both the diagonal and lower left positions, proving the superiority of our method in maintaining label relationships.

# 5  Conclusion

This paper proposes a novel Uncertainty-Aware Hierarchical Refinement scheme for the IIRC task. A global representation extension strategy is presented to enhance the discrimination of incremental classes, and the tricky distillation process is refined with a hierarchical distribution alignment strategy. Consequently, our method involves a multi-level semantic scenery in incremental learning. Experimental results show the superiority of our method in both stability and plasticity.

## Acknowledgments and Disclosure of Funding

Supported by the National Key R&D Program of China under Grant 2020AAA0105701, the National Natural Science Foundation of China (NSFC) under Grants 61872327, and the Major Special Science and Technology Project of Anhui (No. 012223665049). This work was supported by Ant Group through Ant Research Intern Program.

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
