# OpenReview forum: "Uncertainty-Aware Hierarchical Refinement for Incremental Implicitly-Refined Classification"
_NeurIPS.cc/2022/Conference — NeurIPS 2022 Accept_

### Official Review · Reviewer_cq21 · 2022-07-09

**Rating:** 5
**Confidence:** 2
**Soundness:** 3 good
**Presentation:** 3 good
**Contribution:** 3 good

**Summary:**

This paper tackles the Incremental Implicitly-refined classification by proposing a framework that is based on Uncertainty-Aware Hierarchical Refinement. In different stages, label relationships are better leveraged by the proposed method for semantic reasoning, and the uncertainty further assists this inference process. The proposed method achieves competitive performance in two popular benchmarks.

**Questions:**

My major concern is about the incremental novelty of this method. In the few-shot learning community, both class hierarchy with global class representations and the entropy-assisted class margin control have been well exploited.

From the reviewer's perspective, this paper should properly discuss these related methods with necessary statistical comparison.

**Limitations:**

The limitations have been discussed in the appendix.

**Strengths And Weaknesses:**

Strength:

+ Good paper writing and motivation to construct the class hierarchy.

+ Clear illustrations in Figure 4 and Figure 5 that facilitate understanding.

Weaknesses are mainly related to the novelty:

- The hierarchical class structure has been adopted in[1].

- The use of global representation has also been leveraged in [2].

- Entropy-based margin controller has been proposed in [3].

[1] Large-Scale Few-Shot Learning: Knowledge Transfer With Class Hierarchy. CVPR 2019
[2] Few-Shot Learning With Global Class Representations. ICCV 2019
[3] Spot and Learn: A Maximum-Entropy Patch Sampler for Few-Shot Image Classification. CVPR 2019

---

> ### Author Response · Authors · 2022-08-02
> **Thank you for the detailed feedback! We have substantially revised the manuscript as suggested. (Part 2)**
>
> **Q3: Entropy-based margin controller has been proposed in [3].**
>
> A3: Different from [3], which directly utilized the entropy-based uncertainty of samples to guide the control direction of reinforcement learning, we further utilized the class-specific statistical properties of entropy to identify complex semantic relations. In fact, what plays an important role is not the act of employing entropy-based information, but the act of mining the entropy properties in the hierarchical relations to mitigate the subclass-superclass conflict during the incremental process, which is the core of our HDA component. To show the superiority of our method, we conduct an experimental comparison with the entropy-based method utilized in [3]. The experimental results show our HDA component can better maintain hierarchical relationships, and the details are shown below.
>
> | IIRC-Cifar：|   phase 0  | phase 5  | phase 10  | phase 15 |  phase 21 |
> | ---- | ---- |---- |---- |---- |---- |
> |Entropy-based：|    76.26   |  26.64  |   22.20  |    20.28 |   16.75|
> |Ours：      |      77.53   |   30.11  |    25.31  |      23.56  |   19.05 |
>
> [1] Large-Scale Few-Shot Learning: Knowledge Transfer With Class Hierarchy. CVPR 2019
>
> [2] Few-Shot Learning With Global Class Representations. ICCV 2019
>
> [3] Spot and Learn: A Maximum-Entropy Patch Sampler for Few-Shot Image Classification. CVPR 2019
>
> [4] Chen J ,  Wang P ,  Liu J , et al. Label Relation Graphs Enhanced Hierarchical Residual Network for Hierarchical Multi-Granularity Classification[J]. arXiv e-prints, 2022.
>
> [5] Guo Y ,  Xu M ,  Li J , et al. HCSC: Hierarchical Contrastive Selective Coding[J].  2022.
>
> [6] Krishnan S ,  Garg A ,  Liaw R , et al. HIRL: Hierarchical Inverse Reinforcement Learning for Long-Horizon Tasks with Delayed Rewards[J].  2016.
>
> [7] D Kulić,  Nakamura Y . Incremental learning of human behaviors using hierarchical hidden Markov models[C] Intelligent Robots and Systems (IROS), 2010 IEEE/RSJ International Conference on. IEEE, 2010.
>
> [8]Global, Unified Representation of Heterogenous Robot Dynamics Using Composition Operators, 2022,  arXiv:2208.00175
>
> [9] Heterogeneous Anomaly Detection for Software Systems via Attentive Multi-modal Learning, 2022, arXiv:2207.02918
>
> [10] Wang Y ,  Li B ,  Che T , et al. Energy-Based Open-World Uncertainty Modeling for Confidence Calibration[C], 2021.
>
> [11] Chan R ,  Rottman M ,  Gottschalk H . Entropy Maximization and Meta Classification for Out-Of-Distribution Detection in Semantic Segmentation[C]// 2020.

---

> > ### Comment · Reviewer_cq21 · 2022-08-07
> > **The authors' responese are appreciated**
> >
> > Thank you for providing detailed responses. My concerns are addressed, and I will keep my positive rating.

---

> > > ### Author Response · Authors · 2022-08-08
> > > **Thank you for the comment.**
> > >
> > > Dear reviewer,
> > >
> > > We really appreciate your constructive review and your precious time. If you have any further questions or suggestions, we are very happy to discuss with you.

---

> ### Author Response · Authors · 2022-08-02
> **Thank you for the detailed feedback! We have substantially revised the manuscript as suggested. (Part 1)**
>
> We sincerely appreciate your thoughtful comments, efforts, and time. We respond to each of your questions and concerns one-by-one as follows:
>
> **Overall: My major concern is about the incremental novelty of this method. In the few-shot learning community, both class hierarchy with global class representations and the entropy-assisted class margin control have been well exploited. From the reviewer's perspective, this paper should properly discuss these related methods with necessary statistical comparison.**
>
> A: We believe that class hierarchical [1,4,5,6,7], global representations [2,8,9], and entropy-based margin controllers [3,10,11] are widely used in various domains including few-shot and incremental learning. However, we argue that the corresponding definitions, specific implementations, and functions in our paper are entirely different, so that our method is not incrementally innovative. Following your suggestions, we conduct some statistical comparisons with the provided work [1,2,3] to further demonstrate the superiority of our method. We have added corresponding descriptions and experiments in the revised supplementary materials (A.5).
>
> **Q1: The hierarchical class structure has been adopted in [1].**
>
> A1: Our work is different from [1] in both form and content. Firstly, different from the static hierarchical structure that [1] builds by introducing the cross-modal information from text model skip-gram, our method dynamically builds hierarchical structures as the incremental process advances, which relies purely on semantic relationships at the visual feature level.
>
> Secondly, there are dynamically changing and complex subclass-superclass relationships in the IIRC setting, which is difficult to match and adapt by the cross-modal model in [1]. Instead, we consider using the statistical properties of the visual feature distribution itself to adaptively adjust the hierarchical structure, which can better guide the optimization of the implicitly incremental process. To demonstrate it, we try to introduce language models like skip-gram or clip into the IIRC setting, which eventually performed poorly due to their inability to accommodate complex hierarchical relationships, especially in the ImageNet dataset.
>
> Thank you very much for your thoughts, and we will leave how to better facilitate the cross-modal guidance in IIRC setting as our future work.
>
> **Q2: The use of global representation has also been leveraged in [2].**
>
> A2: Our work is different from [2] in both form and content. Firstly, different from [2], in which the global representation is defined as the class-specific representation center from all samples (i.e., prototypes), the global representations mentioned in our paper represents the temporary clustering centers based on the class-specific samples in the fed mini-batch.
>
> Secondly, [2] directly performs regular classification optimization with global representation centers to enhance the discriminability of new classes. Instead, we utilize the distance between the sample and the global representation as a measure of hierarchical importance, to adaptively refine the representation boundaries of the relevant classes.
>
> To demonstrate the superiority of our method, we try to replace our global representation with the one in [2] (i.e., Global_Repre), and find that the performance exhibits a significant degradation due to the inability to adapt to unknown hierarchical relations. The specific experimental results are stated as follows.
>
> |  IIRC-Cifar：|    phase 0 |    phase 5 |    phase 10 |     phase 15  |   phase 21 |
> |  ---- |   ---- |   ---- |   ---- |   ---- |   ---- |
> |  Global_Repre：|   76.62 |      25.42  |    20.38 |       13.66 |      10.80|
> |  Ours：     |     77.53 |       30.11 |      25.31 |        23.56  |      19.05 |

---

### Official Review · Reviewer_o6FB · 2022-07-11

**Rating:** 5
**Confidence:** 4
**Soundness:** 3 good
**Presentation:** 2 fair
**Contribution:** 3 good

**Summary:**

This paper proposes a method called UAHR to address the task of incremental implicitly refinement classification. Specifically, two components, Global Representation Extension (GRE) and Hierarchical Distribution Alignment (HDA) are proposed and integrated together to measure the uncertainty and align the distribution. Experimental studies on public datasets including IIRC-CIFAR and IIRC-ImageNet-lite show the effectiveness of the proposed UAHR scheme.

**Questions:**

While the paper is well motivated and the ablation study is extensive. There are still several issues with this paper.

1, experiments issues

a, in table 1 of the ablation study, in phase 0, why the results of variants with additional one or two of the proposed components are worse than the results of the plain baseline? At the same time, adding GRE seems always harms the performance? How to interpret it?

b, again, in subfigure c of fig. 5,  it seems the major improvement is from the refine component, and the DIV can hardly improve the final performance. This needs explanations.

c, the comparison with SOTA methods is not sufficient. There should be quantitative results (not shown in figure).

d, as subfigures a and b show, the results achieved by the proposed method are not strong, worse than several SOTA methods.

2, missing details

a, in equation 5, why $l_{cls}$ and $l_{dis}$ share the same weight? How is $\gamma$ determined?

b, for the settings of ImageNet, both reference 18 and the following BMVC paper adopt the setting of 10 super classes and 100 subclasses, why do not you follow them to make a fair comparison?

[ref a] HCV: Hierarchy-Consistency Verification for Incremental Implicitly-Refined Classification

3, poor representation

a, there are many long sentences in the paper, making the paper very hard to follow. And there are many grammar issues. I will list some in the following, but they are definitely not all.

b, the figures and tables in the submission should be self-contained. Each equation should be followed by a period or comma.

c, minor: caption in fig. 2, "The class with inheritance relationships have consistency", L169, "we flexibly adjusting ", L175, "And then making the number divide by the number of ...", L176, "The label that final result is 1 has a ...", caption in fig. 4, "The upper row shows the effect of extend the ..."

**Limitations:**

As indicated in the above sections, there are issues with experimental studies, representations, and missing details. For details, please see the above sections.

**Strengths And Weaknesses:**

There are several strengths of the proposed UAHR.

+ The paper is well motivated.
+ The ablation study is extensive.
+ There are many intuitive visualizations of the intermediate results, enhancing the credit of the motivation.

Also, some weaknesses exist.

- The experimental comparison with the SOTA methods is not sufficient.
- The representation of the paper is poor.
- Details are missing.

---

> ### Author Response · Authors · 2022-08-02
> **Thank you for the detailed feedback! We have substantially revised the manuscript as suggested. (Part 2)**
>
> **Q1-d & Q2-b：As subfigures (a) and (b) show, the results achieved by the proposed method are not strong, worse than several SOTA methods. for the settings of ImageNet, both reference 18 and the following BMVC paper adopt the setting of 10 super classes and 100 sub classes, why do not you follow them to make a fair comparison?**
>
> A1-d & A2-b: The top two curves in subfigures (a) and (b) of Fig. 5 represent the upper bound in the IIRC task, which are not SOTA methods. Specifically, incremental joint model has access to the data from all tasks in a complete information setup. Experience Replay that finetunes the model on new classes with infinite buffer to save older samples (ER-infinite) is similar to incremental joint model in an incomplete setup.
>
> Following your suggestion, we have added ImageNet-full [1] and IIRC-ImageNet-Subset [2] for comparison experiments and both obtained SOTA results. We have added the corresponding results (Figure 5 (E) and Figure 5 (F)), the description and citation of HCV in the revised paper.
>
> The results of our experiments are shown below.
>
> | IIRC-ImageNet-Subset |  phase 0   |   phase 2 |     phase 4   |   phase 6   |    phase8 |     phase 10|
> | ----  |  ----  |  ----  |  ----  |  ----  |  ----  |  ----  |
> | Ours：     |        85.25    |   35.60     |   26.94   |    21.27  |       21.58  |     21.09  |
> | iCaRL-CNN：|        84.60 |      32.47  |       23.27  |     17.55  |      16.69 |       16.03 |
> | HCV-iCaRL-CNN： |  85.23   |     40.58    |    29.49    |   22.61  |      13.96   |     12.63|
> | HCV-lucir：     |     84.56  |     38.22   |      22.99  |     15.23    |    12.82|        12.56|
>
> | IIRC-ImageNet-Full |    phase 0 | phase 10  |phase 20|
> | ----  |  ----  |  ----  |  ----  |
> | Ours：     |    74.17 |   17.40  |   14.70 |
> | iCaRL-Norm：|   76.00 |   16.60 |    12.70 |
>
> [1] Mohamed Abdelsalam, Mojtaba Faramarzi, Shagun Sodhani, and Sarath Chandar. IIRC: Incremental Implicitly-Refined Classification. In CVPR, 2021.
>
> [2] Wang K, Liu X, Herranz L, et al. HCV: Hierarchy-Consistency Verification for Incremental Implicitly-Refined Classification. In BMVC. 2021.
>
> **Q2: Missing details.**
>
> **Q2-a: In equation 5, why $l_{cls}$ and $l_{dis}$ share the same weight? How is γ determined?**
>
> A2-a:  It is a common practice to keep the values of classification loss and distillation loss at 1 to 1 in incremental learning, which will facilitate a better dynamic balance [1, 2]. In our work, since $l_{cls}$ and $ l_{dis}$ represents the same BCEwithLogitsLoss, in which the outputs are of the same magnitude, we set their ratio to 1:1 for close loss values. Similarly, to maintain $l_{div}$ in the same magnitude as two above losses, we set the value of γas output ratio between BCEwithLogitsLoss and $l_{div}$. In our paper, γ is set as 10.0 in IIRC-CIFAR and 5.0 in IIRC-ImageNet-Full. To demonstrate the sensitivity of the γ on the final result, we performed experiments in B.2 of supplementary materials. The results show that the value of γ set between 2.5 and 12.5 exhibits a good performance, demonstrating the robustness of this hyperparameter. We have added and modified corresponding description in the revised paper.
>
> [1] Sylvestre-Alvise Rebuffi, Alexander Kolesnikov, Georg Sperl, and Christoph H Lampert. icarl: Incremental classifier and representation learning. In CVPR, 2017.
>
> [2] David Rolnick, Arun Ahuja, Jonathan Schwarz, Timothy Lillicrap, and Gregory Wayne. Experience replay for continual learning. In NeurIPS, 2019.
>
> **Q3: Poor representation.**
>
> A3: Thanks for your careful reading and revision suggestions. We have fixed the mistakes accordingly in the revised paper. We will proofread the draft more carefully as well.

---

> > ### Author Response · Authors · 2022-08-08
> > **We sincerely hope response from the reviewer before the end of the author-reviewer discussion period.**
> >
> > Dear Reviewer o6FB,
> >
> > The author-reviewer discussion period is about to end. Do our responses resolve your initial concerns, and are there any further questions regarding this paper and our new response? We sincerely hope that you can freely share your opinions and suggestions with us and engage with us in the discussion, which is very important for us. We want to thank you again for your time and efforts in improving this work!
> >
> > We are looking forward to hearing from you.

---

> ### Author Response · Authors · 2022-08-02
> **Thank you for the detailed feedback! We have substantially revised the manuscript as suggested. (Part 1)**
>
> We sincerely appreciate your thoughtful comments, efforts, and time. We respond to each of your questions and concerns one-by-one as follows:
>
> **Q1: Experiments issues.**
>
> **Q1-a & b: In table 1 of the ablation study, in phase 0, why the results of variants with an additional one or two of the proposed components are worse than the results of the plain baseline? At the same time, adding GRE seems always harms the performance? How to interpret it? It seems the major improvement is from the refine component, and the DIV can hardly improve the final performance. This needs explanations.**
>
> A1-a & b: For a fair comparison, the same hyperparameter settings (i.e., learning rate 1.0 and 140 training epochs) are adopted in our four ablation experiments. However, GRE poses difficulties for the initial representation optimization (i.e., phase 0) due to the margin constraint on the classification boundary. In this case, the initial performance of GRE is significantly lower than that of baseline by 3 points, and the final performance is comparable after 21 phases. The intention of why we make no special adjustments to this was to demonstrate the superiority of our method by the performance drop from phase 0 to 21. To eliminate the confusion caused by above results and further verify the superiority of GRE, we follow some CIL work [1][2] that adjusts the initial performance for convenient comparison. Specifically, we increase the training epochs to 160 and turn down the learning rate to 0.5, so the initial performance of GRE (i.e., 77.04) is comparable with that of baseline (i.e., 78.53). Under this condition, GRE is obviously superior to Baseline. The new experimental results are shown below:
>
> | IIRC-CIFAR | phase 0   |  phase 5   |  phase 10   |  phase 15   | phase 21|
> |  ----     | ----     |----     |----     |----     |----     |
> | BaseLine：| 78.35   |   26.48    |   21.27 |      18.81    |   17.78     |
> |GRE：  |   77.04   |   26.75  |     21.46  |     19.38    |   18.32   |
> |HDA： |    77.06  |    29.31   |    24.73  |     22.42  |     18.38|
> |GRE+HDA：|77.53   |   30.11    |   25.31   |    23.56   |    19.05 |
>
> As shown in the latest ablation table, as the incremental phase increases, the strengths of GRE in enhancing the representation extensibility of subclasses are gradually emerging, demonstrating its effectiveness. Besides, the DIV mentioned in the question is indeed the GRE, which is a labeling mistake in figures. We have updated the latest ablation experiments and analysis in the revised version.
>
> [1] Zhu K, Zhai W, Cao Y, et al. Self-Sustaining Representation Expansion for Non-Exemplar Class-Incremental Learning. In CVPR. 2022.
>
> [2] Zhang C, Song N, Lin G, et al. Few-shot incremental learning with continually evolved classifiers. In CVPR. 2021.
>
> **Q1-c: The comparison with SOTA methods is not sufficient. There should be quantitative results (not shown in figure).**
>
> A1-c: As shown in the supplementary materials, we show the detailed values of incremental accuracy and other performance indicators (e.g., performance dropping rate [1,2]). Specifically, as shown in Table 8 of supplementary materials, our method has a lower performance dropping rate in the long-range incremental setting, exhibiting a stronger ability to maintain and distinguish hierarchical semantic relationships. As shown in Table 4, our method exhibits a better classification performance on new classes, indicating that the subclass inheritance property is substantially superior to other methods. We have added the corresponding description to remind the reader in the revised paper.
>
> [1] Shi G, Chen J, Zhang W, et al. Overcoming catastrophic forgetting in incremental few-shot learning by finding flat minima. In NeurIPS, 2021.
>
> [2] Zhang C, Song N, Lin G, et al. Few-shot incremental learning with continually evolved classifiers. In CVPR, 2021.

---

### Official Review · Reviewer_qkcE · 2022-07-11

**Rating:** 4
**Confidence:** 3
**Soundness:** 2 fair
**Presentation:** 3 good
**Contribution:** 2 fair

**Summary:**

Inspired by recent Incremental Implicitly-Refined Classification (IIRC) works, this paper proposes an Uncertainty-Aware Hierarchical Refinement (UAHR) method, which explores the semantic correlation of different granularity levels of image labels to ensure the consistency assumption of the feature distribution of class from different granularity levels. Technically, a global representation extension strategy is developed to boost the inter-class discriminability of newly-added classes, and a hierarchical distribution alignment strategy is designed to ensure distribution consistency across different classes belonging to the same super-class. Experiments are conducted on IIRC-CIFAR and IIRC-ImageNet-lite datasets.

**Questions:**

**Major Issues**
>+ Why the RBF kernel can effectively measure the uncertainty of inter-class features.
>+ What are the differences between the KD method used in this paper and the commonly-used KD methods during the rebuttal period.
>+ Inconsistency of experimental results in Figure 5.

**Minor Comments**
>+ In Eq. 1, the definition of K is missing.
>+ Some typos, e.g., Line 145, Page 5. The mathematical symbols j should be edited as the math format rather than the text format.
>+ In Line 270, Page 9, the citation number of SOTA methods (such as ER, AGEM, LUCIR) is missing.


**Limitations:**

Technical contribution: It is interesting that the authors find that, the current incremental training phase would destroy the feature distribution consistency between super- and sub-classes. However, the part about the global representation extension strategy in the paper is relatively weak, due to the following two reasons:
- the authors did not explain why the RBF kernel can effectively measure the uncertainty of inter-class features.
- The knowledge distillation (KD) loss between the current model and old model is a widely-used technique in the class-incremental learning field [Ref 1, 2, 3]. I feel that the author seems to directly apply the KD loss to solve the knowledge forgetting issue, and the KD loss is not adapted according to the IIRC task, which reduces my rating of technical contribution for this paper. Please explain the differences between the KD method used in this paper and the commonly-used KD methods during the rebuttal period.

Technical details:
- The authors use the RBF kernel to calculate the uncertainty between different classes. Why choose the RBF kernel to calculate inter-class uncertainty? Will the effectiveness of the method decline without the RBF mapping.
- Actually, Eq. 1 presents that the mean diversity across all novel class-centers is formulated. But different classes may have different importance for IIRC tasks. As a result, is a class-wise diversity uncertainty metric better?
- In Line 187, Page 5, the authors state that the output entropy of the old classes needs to be subtracted by a margin distance value. It is unclear how to obtain such a margin distance.

Experiments:
- The work [Ref 4] was also conducted on the IIRC-CIFAR and IIRC-ImageNet-lite datasets. But this paper did not compare with the original paper’s results in the work [Ref 4]. Besides, the definition of the horizontal axis in Figure 5 is also inconsistent with that in Figures 3-5 in [Ref 4]. Please explain the reason for this inconsistency during the rebuttal period.

[Ref 1] Semantic-aware knowledge distillation for few-shot class-incremental learning. CVPR-2021
[Ref 2] Few-shot class-incremental learning via relation knowledge distillation. AAAI-2021
[Ref 3] Class-Incremental Learning by Knowledge Distillation with Adaptive Feature Consolidation. CVPR-2022
[Ref 4] IIRC: Incremental Implicitly-Refined Classification. CVPR-2021


**Strengths And Weaknesses:**

1. This paper finds the feature distribution consistency across the classes with inheritance relationships. Further, to guide the optimization of the incremental learning process, this paper designs the hierarchical distribution alignment strategy to exploit the correlation of hierarchical class distributions.
2. Ablation studies show the effectiveness of each proposed module.
3. This paper is well-written, clearly presenting the main idea.

---

> ### Author Response · Authors · 2022-08-02
> **Thank you for the detailed feedback! We have substantially revised the manuscript as suggested.  (Part 2)**
>
> **Q3：Inconsistency of experimental results in Figure 5. Besides, the definition of the horizontal axis in Figure 5 is also inconsistent with that in Figures 3-5 in [Ref 4].**
>
> A3：( 1 ) For a fair comparison, we reproduced the experimental results in Figure 5 with the source code of IIRC to ensure the same setting. Eventually, we found that our results are not exactly consistent with those in the IIRC paper, especially on the IIRC-Cifar dataset. We apologize for not marking the corresponding methods with an asterisk and explaining it. We have updated it in the revised version.
>
> ( 2 ) The definition of the horizontal axis in Figure 5 of our paper is consistent with that in Figures 3-5 in IIRC. Since all the results are below 0.8, we set the upper limit to 0.8 instead of 1 in IIRC for aesthetics. We are sorry that we did not label the indicators of the horizontal, which is actually the same PW-JS metrics as IIRC. We have updated it in the revised version.
>
> **Q4：Actually, Eq. 1 presents that the mean diversity across all novel class-centers is formulated. But different classes may have different importance for IIRC tasks. As a result, is a class-wise diversity uncertainty metric better?**
>
> A4：In fact, Equation 1 represents that the distribution distance between the nearest class and each class in the feature space which acts as the importance of each novel class is utilized to adaptively adjust the corresponding optimization coefficients, which is consistent with the  meaning of class-wise diversity uncertainty metric. Our expression and explanation on this equation is not clear enough, and we have revised it as follows:
>
> $l _{div} = \sum _{c=0}^{n _b} K(h _{\theta}(x) _c, h _{ \theta }(x) _{j _{near}})=\sum _{c=0} ^{n_b} exp \left[
> \frac{-\frac {1}{n_d}\|h _{\theta}(x)_c - h _{\theta}(x) _{j _{near}}\|^2_2}{2\sigma^2}
> \right],$
>
> **Q5：In Line 187, Page 5, the authors state that the output entropy of the old classes needs to be subtracted by a margin distance value. It is unclear how to obtain such a margin distance.**
>
> A5：Following [7], we used a fixed margin value as our hyperparameter when correcting the inter-hierarchical uncertainty based on output entropy. As explained in the main text ( line 163-171 section 3.3 ), although its absolute value is fixed, it adaptively increases or decreases the margin of the old class distribution according to the hierarchical relationship, thus guiding the alignment and discrimination of hierarchical features in the incremental process.
>
> What matters here is its positivity and negativity, not the value. Our further analysis shows that it works well when taking a small value (e.g., 0.1), which would facilitate the guidance of superclasses. To demonstrate its robustness, we performed a set of experiments in B.2 of supplementary materials, and experimental results shows that the value between 0 and 0.25 is a suitable margin interval.
>
> [Ref 1] Cheraghian A, Rahman S, Fang P, et al. Semantic-aware knowledge distillation for few-shot class-incremental learning[C]//Proceedings of the IEEE/CVF Conference on Computer Vision and Pattern Recognition. 2021: 2534-2543.
> [Ref 2] Dong S, Hong X, Tao X, et al. Few-shot class-incremental learning via relation knowledge distillation[C]//Proceedings of the AAAI Conference on Artificial Intelligence. 2021, 35(2): 1255-1263.
> [Ref 3] Kang M, Park J, Han B. Class-Incremental Learning by Knowledge Distillation with Adaptive Feature Consolidation[C]//Proceedings of the IEEE/CVF Conference on Computer Vision and Pattern Recognition. 2022: 16071-16080.
> [Ref 4]Van Amersfoort J, Smith L, Teh Y W, et al. Uncertainty estimation using a single deep deterministic neural network[C]//International conference on machine learning. PMLR, 2020: 9690-9700.
> [Ref 5]LeCun, Y., Bottou, L., Bengio, Y., Haffner, P., et al. Gradient-based learning applied to document recognition. Proceedings of the IEEE, 86(11):2278–2324, 1998a
> [Ref 6]Chen W, Shen Y, Jin H, et al. A variational dirichlet framework for out-of-distribution detection[J]. arXiv preprint arXiv:1811.07308, 2018.
> [Ref 7]Vyas A, Jammalamadaka N, Zhu X, et al. Out-of-distribution detection using an ensemble of self supervised leave-out classifiers[C]//Proceedings of the European Conference on Computer Vision (ECCV). 2018: 550-564.

---

> > ### Author Response · Authors · 2022-08-08
> > **We sincerely hope response from the reviewer before the end of the author-reviewer discussion period.**
> >
> > Dear Reviewer qkcE,
> >
> > The author-reviewer discussion period is coming to an end. Did our response address your initial concerns and do you have any additional questions about this paper? We sincerely hope that you will feel free to engage with us in a discussion by offering your comments and suggestions, which are very important to us. We would like to thank you again for your time and efforts to improve this work!
> >
> > We are looking forward to your reply.

---

> ### Author Response · Authors · 2022-08-02
> **Thank you for the detailed feedback! We have substantially revised the manuscript as suggested. (Part 1)**
>
> We sincerely appreciate your thoughtful comments, efforts, and time. We respond to each of your questions and concerns one-by-one as follows:
>
> **Q1：Why the RBF kernel can effectively measure the uncertainty of inter-class features. Why choose the RBF kernel to calculate inter-class uncertainty? Will the effectiveness of the method decline without the RBF mapping.**
>
> A1：( 1 ) RBF is a commonly used method to estimate the sample uncertainty based on the output logits [4, 5], which is similar to the entropy-based one [6]. The difference is that RBF nonlinearly maps the logits to a high-dimensional space and calculates the distance relationship with the all class means, which serves as the estimate of inter-class uncertainty.
>
> ( 2 ) In IIRC setting, there is a magnitude gap among the hierarchical output logits due to the complex subclass-superclass relationship during the incremental process. To solve this problem, we apply the RBF kernel function to adjust the logits adaptively before the class-specific uncertainty estimation so that they can be reasonably compared.
>
> ( 3 ) We conduct a comparison experiment between the methods with and without (i.e., simple-dis) RBF mapping, verifying its effectiveness from incremental performance. The experimental results are shown below, and we have added it (A.5 in the supplementary material) in the revised paper.
>
>
> |  IIRC-Cifar | phase 0  |  phase 1  |  phase 2  | phase 3 |  phase 5  | phase 10  |  phase 15 |  phase 21|
> |  ----    |  ----    |  ----    |  ----    |  ----    |  ----    |  ----    |  ----    |  ----    |
> | ours     |     77.53   |   55.70 |    41.05  |   35.21 |   30.10  |   27.88  |     23.56   |   19.04|
> | simple-dis  |   78.83   |   52.37  |   39.31 |    34.64 |   28.97  |   24.62  |     22.46 |     18.02|
>
> **Q2：What are the differences between the KD method used in this paper and the commonly-used KD methods.**
>
> A2 : For fair comparison, we adopt the same base distillation function with IIRC, which is similar to the distillation function in some CIL methods [1,2,3]. However, as shown in introduction (lines 50-53) and Figure 3, the proposed hierarchical refinement module is the one which matters, not the knowledge distillation itself. Different from [1,2,3] that deal with feature confusion by maintaining semantic relationships among the same hierarchy, our method is proposed to mitigate semantic conflicts across hierarchies to refine the distillation process. In fact, our hierarchical refinement module can benefit from the improvements of the distillation process including [1,2,3] due to it orthogonality to the KD methods. To further demonstrate the difference, we directly replace our KD and hierarchical refinement module with the KD method in [1] for comparison. The experimental results prove the difference and effectiveness of our proposed hierarchical refinement module, which is stated as follows.
>
> | IIRC-Cifar |  phase 0 |  phase 1  | phase 2|  phase 3 | phase 5 | phase 10 |  phase 15 | phase 21  |
> | ---- |---- |---- |---- |---- |---- |---- |---- |---- |
> |ours      |       77.53 |    55.70    |41.05  |  35.21 |  30.10   | 27.88 |     23.56  |   19.04|
> |Semantic-aware  |  76.22  |   49.65  |  40.66  |  33.99 |   25.27  |  21.38   |   13.46  |   10.62|
>
> Specifically, as analyzed in our introduction, the difficulty in maintaining the old knowledge for IIRC setting is how to discriminate the subclass-superclass relationship and correct the optimization conflict during the distillation process. All above methods [1,2,3] focus on the full retention of old knowledge, which essentially cannot solve the problem. In this paper, our HDA aligns superclasses by knowledge distillation while ensuring the separability of subclasses by adjusting the entropy values of the relevant superclass to ensure the consistency of hierarchical uncertainty.

---

> ### Comment · Reviewer_qkcE · 2022-08-09
> **More experimental analyses are necessary**
>
> Thanks for the authors’ detailed response, which addresses my concerns about 1) the motivation of the RBF kernel; 2) the effectiveness of KD. But the authors seem to ignore the reviewer’s major concern about 1) the missing quantitative comparisons with the work [Ref 4], which is a recent representative work in the field of IIRC; 2) a detailed explanation of why this paper’s results reproduced by the authors are not exactly consistent with those in the work [Ref 4]. Thus, it is recommended that the revision needs to supplement more quantitative results for the major dataset such as the IIRC-CIFAR dataset, and add some analyses about why the reproduced results are lower than those in the original paper.
>
> Overall, this paper is well-written and clearly presents the main idea. But more experimental analyses and comprehensive comparisons are necessary. Thus, I decided to keep my rating.
>
> [Ref 4] IIRC: Incremental Implicitly Refined Classification.

---

### Official Review · Reviewer_wtSW · 2022-07-11

**Rating:** 5
**Confidence:** 4
**Soundness:** 3 good
**Presentation:** 3 good
**Contribution:** 2 fair

**Summary:**

This paper proposes an uncertainty-aware hierarchical refinement scheme for incremental implicitly-refined classification, exploring inheritance relations of multi-level semantic increment. It consists of a global representation extension module and a hierarchical distribution alignment module.

**Questions:**

1.	More comparison experiments should be conducted on other datasets such as IIRC-ImageNet-full to further verify the effectiveness of proposed method, which is commonly used in IIRC tasks.
2.	The effectiveness of the GRE module is not verified in ablation studies, according to the results in Table 1. More detailed analyses should be given.
3.	Most part of the performance decrease happens in first 5 phases, focusing on how to maintain more robust performance in the beginning may lead to better results in latter phases.
4.	In Table 2, the common uncertainty method should be introduced briefly, so the comparison can verify the effectiveness of proposed method. Numerical results of comparison experiments should be provided to better illustrate the improvement of proposed method.
5.	More details should be provided as explanation for Figure 6, the present version cannot clearly show the improvement in maintaining label relations.


**Limitations:**

The authors should give more introduction and limitation analyses about the proposed method in realistic application scenarios.

**Strengths And Weaknesses:**

Strengths:

1.	This paper introduces global representation extension and hierarchical distribution alignment to the IIRC task for better hierarchical invariance, and conducts experiments on IIRC-CIFAR dataset to verify its effectiveness.
2.	This paper makes progress in a newly proposed task, IIRC, which focus on the robustness of models when processing data of different categories and different-grained classes.

Weakness:

1.	There is only one dataset for overall performance comparison experiments. More experiments and analyses on more datasets should be added to verify the effectiveness of the proposed method.
2.	The effectiveness of separate module is not fully verified. The ablation results of the GRE module in Table 1 is lower than the baseline, more explanations should be given.

---

> ### Author Response · Authors · 2022-08-02
> **Thank you for the detailed feedback! We have substantially revised the manuscript as suggested.  (Part 2)**
>
> **Q4：In Table 2, the common uncertainty method should be introduced briefly, so the comparison can verify the effectiveness of proposed method. Numerical results of comparison experiments should be provided to better illustrate the improvement of proposed method.**
>
> A4：A detailed version of Table 2 is presented in section A.3 of Supplementary Materials. As described in section A.3, we present and compare more classical uncertainty methods to numerically demonstrate the superiority of our approach. Specifically, the entropy-based uncertainty method ‘mean’ and energy-based uncertainty method ‘std’ is widely used in the OOD [1, 2] and regression task [7], respectively. The uncertainty method ‘cov’ based on the coefficient of variation is widely used in the segmentation domain [5, 6]. We applied these uncertainty methods in the IIRC setting and found suboptimal performance due to the confusion on the hierarchical relationship, illustrating the improvement of our proposed method. We apologize for the confusing presentation and have introduced a brief description in the main manuscript and supplementary materials.
>
> [1] Chen W, Shen Y, Jin H, et al. A variational dirichlet framework for out-of-distribution detection. arXiv preprint arXiv:1811.07308, 2018.
>
> [2] A. Vyas, N. Jammalamadaka, et al. Out-of-distribution detection using an ensemble of self-supervised leave-out classifiers. In ECCV, 2018.
>
> [3] P. McClure, N. Rho, et al. Knowing what you know in brain segmentation using bayesian deep neural networks. Frontiers in neuroinformatics, 2019.
>
> [4] P. Seebock, J. I. Orlando, et al. Exploiting epistemic uncertainty of anatomy segmentation for anomaly detection in retinal oct. TMI, 2020.
>
> [5] A. G. Roy, S. Conjeti, et al. Bayesian quicknat: Model uncertainty in deep whole-brain segmentation for structure-wise quality control.
>  NeuroImage, 2019.
>
> [6] G. Wang, W. Li, et al. Aleatoric uncertainty estimation with test-time augmentation for medical image segmentation with convolutional neural networks. Neurocomputing, 2019.
>
> [7] Levi D, Gispan L, et al. Evaluating and calibrating uncertainty prediction in regression tasks. Sensors, 2022.
>
> **Q5：More details should be provided as explanation for Figure 6, the present version cannot clearly show the improvement in maintaining label relations.**
>
> A5：In the IIRC task, the diagonal values in the confusion matrix represent the prediction of subclass labels (i.e., same-level label relationships), while the bottom-left corner represents one of superclasses. A more similar lower left corner compared to the ground truth represents better label relationship maintenance between subclass and superclass. As shown in Figure 6, our method performs best in both the diagonal and lower left positions, proving the superiority of our approach in maintaining label relationships. We are very sorry for the missing ground truth, and we have added it in the revised version. In addition to Figure 6, Figure 4 is dedicated to visualizing the role of our method in correcting the hierarchical optimization conflict, which further demonstrates the superiority in maintaining label relationships.
>
> **Limitations： The authors should give more introduction and limitation analyses about the proposed method in realistic application scenarios.**
>
> Answer： Our method focuses on representation generalization to other fine-grained classes, which is suitable for application tasks in the case of large model pre-training such as weakly supervised image segmentation. However, in fine-to-coarse application scenarios such as robot self-learning, subclass-superclass recognition without sequential relationships is required. In this case, our method suffers from the weak generalization of the initial representation and the lack of semantic structures due to feature confusion. How to enhance the robustness and introduce language models to better guide the semantic relations should draw wider attention. We have updated the description of the limitations in the revised paper.

---

> > ### Author Response · Authors · 2022-08-08
> > **We sincerely hope response from the reviewer before the end of the author-reviewer discussion period.**
> >
> > Dear Reviewer wtSW,
> >
> > The author-reviewer discussion period is coming to an end. Did our response address your initial concerns and do you have any additional questions about this paper? We sincerely hope that you will feel free to engage with us in a discussion by offering your comments and suggestions, which are very important to us. We would like to thank you again for your time and efforts to improve this work!
> >
> > We are looking forward to your reply.

---

> ### Author Response · Authors · 2022-08-02
> **Thank you for the detailed feedback! We have substantially revised the manuscript as suggested.  (Part 1)**
>
> We sincerely appreciate your thoughtful comments, efforts, and time. We respond to each of your questions and concerns one-by-one as follows:
>
> **Q1：More comparison experiments should be conducted on other datasets such as IIRC-ImageNet-full to further verify the effectiveness of the proposed method, which is commonly used in IIRC tasks.**
>
> A1：We want to thank you for your time and efforts in improving this work! Following your suggestion, we add the experiments on IIRC-ImageNet-Full [1] and IIRC-ImageNet-Subset [2] datasets for further comparison in addition to IIRC-CIFAR and IIRC-ImageNet-Lite datasets in the main manuscript. The proposed method achieves consistent SOTA results on all IIRC benchmarks, demonstrating the better generalization and effectiveness of our model on multiple datasets. We have revised the corresponding contents in the revision paper (Figure 5) and supplementary material (Table 9, Table 10). The specific experimental results are shown as follows:
>
> |     IIRC-ImageNet-Full        | phase 0 | phase 10 | phase 20 |
> | ----         | ----        |    ----         |     ----       |
> | Ours：  |       74.17 |   17.40  |   14.70    |
> |iCaRL-Norm：|   76.00 |     16.60  |   12.70  |
>
> | IIRC-ImageNet-Subset |    phase 0  |  phase 10 |
> | ----         | ----        |    ----         |
> |Ours：  |          85.25 |     21.09   |
> |iCaRL-CNN：   |    84.60   |   16.03  |
> |HCV-iCaRL-CNN： | 85.23 |     12.63 |
>
> [1] Mohamed Abdelsalam, Mojtaba Faramarzi, et al. IIRC: Incremental Implicitly-Refined Classification. In CVPR, 2021.
>
> [2] Wang K, Liu X, Herranz L, et al. HCV: Hierarchy-Consistency Verification for Incremental Implicitly-Refined Classification. In BMVC, 2021.
>
> **Q2：The effectiveness of the GRE module is not verified in ablation studies, according to the results in Table 1. More detailed analyses should be given.**
>
> A2：Thanks for your suggestions. For a fair comparison, the same hyperparameter settings (i.e., the initial learning rate is 1.0, and the training epoch is 140) are adopted in the four ablation experiments. However, GRE poses difficulties for the initial representation optimization (i.e., phase 0) due to the margin constraint on the classification boundary. In this case, the initial performance of GRE is significantly lower than that of baseline by 3 points, and the final performance is comparable after 21 phases. The reason why we make no special adjustments was to demonstrate the superiority of our method by the performance variation from phase 0 to 21. To eliminate the confusion caused by above results and further verify the superiority of GRE, we follow the way of some CIL work [1, 2] that adjusts the initial performance for convenient comparison. Specifically, we increase the training epochs to 160 and turn down the learning rate to 0.5, so the initial performance of GRE (i.e., 77.04) is comparable with that of baseline (i.e., 78.53). Under this condition, GRE is obviously superior to Baseline. The new experimental results are shown below:
>  |IIRC-CIFAR|   phase 0  |   phase 5   |  phase 10  |   phase 15 |   phase 21 |
> |    ----         | ----        |    ----         |      ---- |    ----    |         ---- |
> |Baseline：| 78.35   |   26.48   |    21.27  |     18.81      | 17.78 |
> |GRE：     |77.04    |  26.75    |   21.46     |  19.38       |18.32  |
> |HDA：     |77.06    |  29.31    |   24.73    |   22.42      | 18.38  |
> |GRE+HDA：|77.53  |    30.11  |     25.31  |     23.56 |      19.05  |
>
> As shown in the table above, as the incremental phase increases, the strengths of GRE in enhancing the representation extensibility of subclasses are gradually emerging. We have updated the latest ablation experiments and analysis in the revised version.
>
> [1] Zhu K, et al. Self-Sustaining Representation Expansion for Non-Exemplar Class-Incremental Learning. In CVPR, 2022.
>
> [2] Zhang C, et al. Few-shot incremental learning with continually evolved classifiers. In CVPR, 2021.
>
> **Q3：Most part of the performance decrease happens in first 5 phases, focusing on how to maintain more robust performance in the beginning may lead to better results in latter phases.**
>
> A3：The performance degradation in the initial five phase is mainly caused by subclass confusion and the conflict between subclass and superclass. The former is mainly affected by the initial representation, while the latter is affected by the incremental optimization process. It is undeniable that maintaining more robust performance in the beginning has a positive impact on enhancing subsequent subclass discriminability. Orthogonally, we focus on how to continuously adapt our optimization strategy based on the subclass-superclass relationship during the incremental optimization process, which is equally important to the overall performance. For a fair comparison with the existing methods under similar condition, we do not consider enhancing the initial representation. We will leave it as our future work.

---

### Author Response · Authors · 2022-08-06
**Looking forward to the reviewer's reply !**

Dear reviewers,

Thanks a lot for your time and efforts in reviewing our paper. We have tried our best to address all mentioned concerns. We would appreciate it if you could take a look at our response.  As the discussion deadline is approaching, your feedback is very important to us, and if there are any new questions, we can therefore reply in time.

Sincerely yours,

Authors

---

### Meta-Review · Area_Chair_7K4t · 2022-08-25

**Recommendation:** Accept
**Confidence:** Certain

**Metareview:**

Three reviewers are positive on this paper. Although the rating of one reviewer is borderline reject,  he is fairly confident in his assessment.  In effect,the authors have well addressed all the reviewers' concerns in the rebuttal.  So I suggest accepting this paper.

**Award:**

No

---

### Decision · Program_Chairs · 2022-09-14

Accept